# Gender and Educational Inequalities during the COVID-19 Pandemic: Preliminary Insights from Poland

**Małgorzata Krywult-Albańska [1],\* and Łukasz Albański [2]**

1   Institute of Philosophy and Sociology, Pedagogical University of Cracow, 30-084 Kraków, Poland
2   Institute of Educational Sciences, Pedagogical University of Cracow, 30-060 Kraków, Poland;
    lukasz.albanski@up.krakow.pl
\*   Correspondence: malgorzata.krywult-albanska@up.krakow.pl

**Abstract:** The global pandemic of COVID-19 has had a profound impact on many spheres of social life across the world. One of them has been the deepening of social inequalities and the aggravating of discrimination based on gender. Emerging studies in the field of education and occupation systems point to the fact that women seem to have been particularly affected, along with layoffs in those sectors of the economy where female staffs prevail. Additionally, in many countries, the burden of combining professional careers and supporting the education of young children falls disproportionately on mothers. These transformations pose a challenge to meeting the United Nations Sustainable Development Goals, wherein gender equality is an important factor. This article uses official statistical data to examine gender and educational structures during the COVID-19 pandemic in Poland, set against the backdrop of other European nations and analyzed in the context of sustainability. Have educational and gender inequalities been exacerbated as data from other countries suggest? In order to answer this question, the article traces changes in the education system in Poland and their implications for gender structures. The latter have also been affected by transformations on the labor market in various sectors of the economy, therefore, the second part of the analysis focuses on the labor market changes during the pandemic. The final section offers conclusions on the implications of the pandemic for the studied issues. Throughout the article, we apply the principles of unobtrusive research. Following the theoretical framework outlined in the first part of the paper, we carry out a descriptive analysis of existing statistical data collected by the Eurostat. These official statistics are supplemented by an overview of public opinion polls to allow for perspectives on structural changes, as they are perceived by those affected by them.

**Keywords:** educational inequalities; gender inequalities; COVID-19 pandemic

## 1. Introduction

One of the largest disruptions created by the COVID-19 pandemic occurred in the education system, affecting around 1.6 billion students worldwide. Many schools went online in the spring of 2020 as the virus spread around the world, turning to virtual education to replace in-person learning. According to the UN, closures of schools and other learning spaces have impacted 94 per cent of the world's student population, up to 99 per cent in low and lower-middle income countries, leaving them without in-person learning [1].

On the one hand, the pandemic has stimulated innovation within the education system. Innovative approaches have been implemented in support of education and training continuity, and distance learning solutions were developed. On the other hand, these changes have not been distributed equally across and within countries. Pre-existing educational disparities are being exacerbated by reducing the opportunities of the most vulnerable children, youth and adults (including girls and young women) to continue their learning. Children and youth affected by a lack of resources or an enabling environment

to access learning have been left behind. To mention just one aspect of these inequalities, more than 460 million students across the globe do not have Internet access, computers or mobile devices to participate in virtual learning while their schools are closed [1]. Around 24 additional children and youth (from pre-primary to tertiary levels) were estimated to drop out or not have access to school in 2021 due to the pandemic's economic impact alone.

Across the world, girls in particular are likely to drop out, leaving them vulnerable to child marriage, early pregnancy and domestic violence. These problems are being continually raised by human rights activists and international organizations, pointing to the fact that in the aftermath of the pandemic, progress on gender equality has been set back decades and is part of a "pandemic of human rights abuses" (in the words of the UN secretary general, António Guterres), abuses which "thrived because poverty, discrimination, the destruction of our natural environment and other human rights failures have created enormous fragilities in our societies" [2]. According to the UN secretary general, "The crisis has a woman's face. Violence against women and girls in all forms has skyrocketed, from online abuse to domestic violence, trafficking, sexual exploitation and child marriage." Due to higher drop-out rates for girls, the COVID-19 pandemic threatens gains in education for women.

Meanwhile, numerous studies emerge pointing to yet another gender dimension of the current crisis: Not only are women at a greater risk of contracting COVID-19 as frontline workers overrepresented in the services sector in occupations which cannot switch to remote work, but they also work in industries experiencing more economic distress and are primary caregivers shouldering increased domestic work due to school and daycare closures [3]. It is at this point where changes on the labor market overlap with transformations in the educational and gender structures, each exerting a ripple effect on the others. There is a concern that these transformations might compromise sustainable development goals set out in the 2030 Agenda for Sustainable Development, adopted by the UN General Assembly in September 2015. In particular, they pose a threat to gender equality and reducing of other inequalities (sustainable development goals 5 and 10).

In this research, the focus is on Poland and on the dynamic between the following three areas: educational, economic and gender structures during the COVID-19 pandemic in the context of sustainability. Educational, labor and gender inequalities are studied together because changes in each of these areas exerts ripple effects on the others. Poland is set against the backdrop of other European Union nations. Firstly, an overview is provided of the research conducted so far on the impact of the coronavirus pandemic on the educational, gender and occupational structures in high-income countries from the standpoint of sustainability. Secondly, the article focuses on selected statistical data to study COVID-19 pandemic's impact on educational, gender and economic structures in Poland. Drawing mostly on Eurostat's data and public opinion surveys, we find that increased burden due to care-related needs for children has been cushioned by high inactivity rates for women, preserving existing gender structures.

## 2. Background: Justification for Research on Education, Gender and Sustainability

### 2.1. Educational System and Inequalities

In the 2030 Agenda for Sustainable Development adopted by the UN General Assembly in September 2015, education is a primary driver of progress across all 17 Sustainable Development Goals, "key to sustaining peaceful, prosperous and productive societies" [4]. ('Development which meets the needs of the current generations without compromising the ability of future generations to meet their own needs' (6). This is the definition of sustainable development that was first introduced in the Brundtland report by the World Commission on Environment and Development (WCED) in 1987, and it is the most widely used nowadays.) It is also listed as a separate goal (sustainable development goal 4 (SDG 4): quality education), along with gender equality (SDG 5) and reduced inequalities (SDG 10). Progress within each goal is measured by a set of indicators. The following section reviews potential threats to achieving these goals caused by the coronavirus pandemic, looking at

selected indicators relevant to the subject of this article (Not all indicators seem relevant, e.g., SG 4 also covers, among others, tertiary educational attainment and adult participation in learning which will not be discussed herein.) (Not all indicators seem relevant, e.g., SG 4 covers, among others, also tertiary educational attainment and adult participation in learning which will not be discussed herein.).

Sustainable development goal 4 seeks to ensure inclusive and equitable quality education and promote lifelong learning opportunities for all. This goal, apart from seeking to ensure equitable and quality education through all stages of life, envisages the elimination of gender and income disparities in access to education. Progress within this target is measured with indicators such as participation in early childhood education, the number of early leavers from education and training, as well as underachievement in reading, maths and science. There is evidence from some countries that lockdown measures introduced in the aftermath of the pandemic negatively affected each of these indicators. Learning outcomes of children from families with limited material and human resources have deteriorated, and some children might have dropped out of the educational system altogether. Even before the pandemic began, researchers had pointed to the fact that across the world, students had been increasingly at school but had not been learning, obtaining bad outcomes in terms of learning results [5]. In other words, learning had stagnated while schooling had increased. This was true particularly for lower-income countries and deemed important because human capital measured by learning is associated with growth. Moving education to the virtual realm poses additional challenge, adding yet another concern. Research carried out in some countries suggests that remote instruction has been a poor substitute for in-person schooling for many students, resulting in declining learning time and lower levels of engagement. For example, studies conducted in the UK show that learning time for children declined as the pandemic progressed, particularly for secondary school pupils (research by the Institute for Fiscal Studies tracking the activities of 650 school-aged children in April–May, during full lockdown, and again in June–July when restrictions were eased [6]). Before the pandemic lockdowns, the IFS estimated that secondary school pupils worked for about 6.5 h each day on average. That number slipped to 4 hours and 15 min in April–May and then shrank by a further 50 min by the start of the summer holidays. Simultaneously, the assessment by the National Foundation for Educational Research revealed sharp falls in the rate of progress in mathematics, reading and writing made by primary school pupils. An earlier IFS study also suggested that "lost learning" could account for a EUR 350bn fall in lifetime earnings by those educated during the lockdowns and school closures over the first year after the pandemic's outbreak. When schools reopened, well-off families were far more likely to send their children in to school compared with families from disadvantaged backgrounds—potentially worsening educational inequalities in the UK (Parents reported that their main reason for delaying a return to school was for health concerns, with disadvantaged families saying they were reluctant for their children to be among the first ones to return, as well as citing practical difficulties such as transport.).

In a study by Domina et al. [7] in the US, remote instruction was associated with lower levels of engagement among pupils, varying, however, mostly relative to access to technological resources, such as high-speed Internet and Internet-enabled devices and staying socially connected to other students' families. The latter factor highlights the role of social capital in educational performance. Overall, the authors of the study hypothesize that household material and technological resources, school programming and instructional strategies and family social capital contribute to student engagement in remote learning. It is estimated that 58.1 percent of U.S. children participated in online learning, and of these, 10.1 percent did not have adequate access to both the Internet and a computer or other electronic device used for educational purposes. In a study by Friedman et al. [8], the rates of inadequate access varied nearly 20-fold across the gradient of parental race/ethnicity and education, from 1.9 percent for children of Asian parents with graduate degrees to 35.5 percent among children of Black parents with less than a high school education. As

the authors of the study succinctly summarize: "These findings indicate alarming gaps in potential learning among U.S. children" (Research carried out in Canada estimated that the socio-economic skills gap could increase by more than 30 per cent due to the pandemic [9]). Children expected to participate in online learning without adequate technology are highly unlikely to achieve significant learning compared with their peers to whom more resources are available. Falling behind in learning goals, they may also be more likely to leave school entirely. In the countries of the European Union, access to distance learning varied between near-universal in Slovenia, where fewer than 2% of pupils could not be reached, and countries such as Italy, where 48% of pupils were left without education [10].

These and other studies clearly demonstrate that school closures have had, in general, a negative effect on learning outcomes in many countries but also that these outcomes depend strongly on parents' resources, such as access to modern technologies, digital skills and the time available to spend assisting children learning at home (see also a review of the literature in [9]). Whereas many households sought online learning resources during the pandemic in order to compensate for lost school-based learning time, their engagement with these resources may have been significantly more intense in areas with higher income, better Internet access and fewer rural schools [11]. Such differences will likely widen achievement gaps along these dimensions. This effect is already observable in some countries, where despite an overall surge in the best grades among pupils at the secondary level, the gap between pupils from more affluent and from disadvantaged backgrounds has widened [12–14]. These transformations run counter to sustainable development goal 10 which *addresses inequalities within and among countries. It calls for nations to reduce inequalities in income and those based on age, sex, disability, race, ethnicity, origin, religion or economic or other status within a country. (The goal also addresses inequalities among countries, including those related to representation, and calls for the facilitation of orderly and safe migration and mobility of people.)* Fulfilling this goal seems particularly challenging considering the fact that low-income households may be affected disproportionately by the COVID-19 crisis [15] (p. 137). Inequalities in the long run may thus be exacerbated, with particularly lasting impacts on children [16].

### 2.2. Ripple Effects of Changes in the Educational System

Limited access to early childhood education and care during lockdowns might have prevented parents, especially women, from (re)integrating in the labor market, running counter to sustainable development goal 5, which *aims to achieve gender equality by ending all forms of discrimination, violence and any harmful practices against women and girls in the public and private spheres. (It also calls for the full participation of women and equal opportunities for leadership at all levels of political and economic decision making)* [15] (p. 127). Many scholars have pointed to the fact that the pandemic may exacerbate gender inequalities as women reduced their labor market participation to meet increased caregiving needs.

A study carried out at the beginning of the pandemic in a few European countries demonstrated that 60 per cent of parents in France, Germany, Italy, the UK as well as the USA were unable to find alternative solutions for schools and day-care centres [17]. Considering the fact that the primary responsibility for home and family has traditionally been assigned to women, it may be assumed that women bore the greater share of additional time spent on childcare and household tasks. Meanwhile, economic disruption might contribute to higher earning gaps, thus widening gender inequality. Indeed, there is some evidence that working mothers have experienced greater challenges during the pandemic [18–20]. Qian and Fuller [20] found that Canadian mothers were more likely to become unemployed from February to May 2020, compared with fathers, especially for those whose youngest children were ages 6 to 12. In the USA, the white–non-white gap in employment increased significantly during the post-outbreak period, with results from individual fixed-effects regression models showing a strong white male advantage in the likelihood of being laid off for post outbreak months compared with women, black men, Hispanic men and Asian men [21].

On the other hand, Schieman et al. [22] found no differences between men and women in the patterns of changes in work–life conflict during the pandemic (women did not experience more burden with childcare). Women with younger children were not more likely to stop working or drop out from the survey carried out about nine months from the start of the pandemic (it is not known, though, whether there were differences by the level of education) (The study revealed, however, that although overall levels of work–life conflict decreased in the population during the pandemic, the presence of children at home generated a countervailing force—any decrease in work–life conflict decreased among those with children under 13.). These impacts will likely differ depending on country-specific gender roles and expectations, which have also been evolving under lockdown conditions. For example, a recent study of Canadian parents during the pandemic suggests slightly more egalitarian work–home and family-care arrangements between women and men [23], while a study by Mize et al. [24] revealed a shift toward more conventional gender parenting attitudes in the US.

*2.3. Labour Market Transformations*

The pandemic and associated mitigation measures have brought three major changes to the labour market and work institutions [25]: (1) increased unemployment rates and organizational downsizing; (2) amplified health risks and challenges for essential workers who are required to work on-site; and (3) unprecedented shifts to telecommuting. The gendered impacts of these changes have already been documented, while some areas remain yet to be studied (see [25] for a review of the situation in the US). In many cases, changes in the labor market and systems of care have deepened pre-existing gender inequalities in both areas of work and family. Adams-Prassl et al. [26] found women to be significantly more likely to lose their jobs in the aftermath of the pandemic's breakout in the United States and United Kingdom but not in Germany. They also observed that women who did not lose their job were no more likely to experience a fall in their income compared to men in all three countries. In a study carried out in April in six countries in Europe, Asia and the US, Dang and Nguyen [27] found that although no gender differences existed in the COVID-19 impacts on temporary job loss, women were 24 percent more likely to permanently lose their job compared to men. Women also tended to reduce their consumption and increase savings more than men, probably due to concerns about the future effect of the pandemic on their labor income. These differences could be explained by the higher rate of working in services jobs for women than men (a 10 to 20 percentage point gender gap for the countries covered by the survey) and the service sector was more affected by COVID-19 than other sectors of the economy [28]. The authors of the study found heterogenous effects on women across countries, likely due to different COVID-19 infection rates and the shares of women participating in the labor force.

Should the negative economic impacts of the pandemic (in terms of the probability of job loss and lost income) add up to the ripple effects of the disruption of the education system, it might be expected that the probability of aggravating existing inequalities along gender and class lines will rise. Most studies referred to above imply that this indeed might be the case in some contexts and that the deepening of social inequalities is highly likely, threatening the achievement of UN's sustainable development goals. Is Poland one of those contexts? The following section focuses on the situation in this country, set against the backdrop of other European nations.

## 3. Research Questions

In this research, the focus is on Poland and on the dynamic between the three areas: the educational, gender and economic structures during the COVID-19 pandemic in the context of sustainability. We set this country against the backdrop of other European nations. In particular, we aim to address the following questions:

1. In what way has the pandemic affected the education system in Poland? In particular, is there evidence that the pandemic has deepened educational inequalities, as

      elsewhere across the world? Is it likely that COVID-19 will exacerbate existing gaps in schooling?

2.    How has the pandemic affected women's and men's position on the labor market? The COVID-19 pandemic strongly affects sectors with high shares of female employment. Another channel of impacts of COVID-19 is an increased need for childcare for mothers due to school closures. It might therefore be hypothesized that the pandemic has affected women's situation on the labor market in a more negative way than men's situation. Moreover, it might be hypothesized that households' resources (their economic, social and cultural capital) have been negatively affected during the pandemic, potentially contributing to gaps in schooling mentioned in point 1 above.

3.    In what way has the pandemic affected gender and parental roles in Poland? In particular, have changes occurred in social attitudes with the potential to ease the burden that the pandemic has placed on children's schooling? A US study [24] shows that there has also been a shift in attitudes about parenting roles. Is this also true for Poland? Has progress in gender equity been reversed, or is there evidence to the contrary?

      While answering the aforementioned questions, we aim to elucidate the multiple ways in which the effects of the pandemic in educational, labor and gender structures overlap in the context of sustainability.

## 4. Methods

      In this research, we use data from multiple sources to examine educational, labor and gender inequalities during the COVID-19 pandemic in Poland, set against the backdrop of other European countries and viewed from the standpoint of sustainability. Throughout the analysis, we apply the principles of unobtrusive research. Following the theoretical framework outlined in the preceding paragraphs, we carry out analysis of existing statistical data collected by the Eurostat. These official statistics are supplemented by an overview of public opinion polls (by Public Opinion Research Center (CBOS) in Poland), to allow for perspectives on structural changes, as they are perceived by those affected by them.

      While examining Eurostat's data, we draw on the most recent data available (usually for the year 2020), juxtaposing them with preceding years to capture any changes that might have occurred in the period of time before and after the COVID-19 pandemic's outbreak. In view of the focus of our research, we analyze statistics which are indicative of progress within sustainable development goals (as stipulated in the preceding paragraphs).

      The analysis herein comes with the caveat that the pandemic is an ongoing process, with new variants of the virus emerging in various places around the world and preventive measures being reintroduced. This article provides preliminary insights into educational, labor and gender-related impacts as of the time of writing.

      Some of our research questions (especially the third one, about the pandemic's effect on gender and parental roles) cannot be answered based on official statistics but seem important to include in order to provide a comprehensive picture of the studied phenomena. Therefore, apart from the Eurostat's data, we analyze representative public opinion polls, looking for patterns of responses which might be indicative of the pandemic's social impacts.

      Occasionally, we draw on research results published by other sources. Our research is descriptive. We provide an overview of statistics which might serve as a background for more in-depth studies.

## 5. Results

### 5.1. Background Indicators on Digitalization

      In Poland, the same as elsewhere throughout the world, the pandemic has transformed children's educational experiences. In an effort to contain the virus, schools around the country were closed and a months-long effort was made to provide remote education for homebound youth. In most cases, these students' schools replaced in-person

instruction with a mix of synchronous and asynchronous instruction offered via Web-based instructional technologies such as Microsoft Teams and Zoom. Therefore, access to these technologies is what should be examined in the first place in order to assess the pandemic's impact on the educational system in Poland.

In 2020, 60% of households in Poland had a fixed, very-high-capacity network (VHCN) connection (equal to the average share in the EU) [15] (p. 205). However, less than 20% of rural households enjoyed such connection [29]. These data point to the disadvantaged position of rural areas in Poland, potentially influencing scholarly performance of pupils living on these areas. In terms of connectivity, Poland does not score the highest in Europe on the Digital Economy and Society Index. Its score of around 51 (covering components such as fixed broadband take-up, fixed broadband coverage and mobile broadband) places it behind European leaders: Denmark, Sweden and Luxemburg (score over 60), around the middle of the ranking. Poland, along with Lithuania, Romania and Slovakia, also lags behind other EU countries in terms of fixed coverage, with less than 90% of households covered. The coverage of next generation access (NGA) technologies is particularly low in Poland, standing at around 75% in urban areas and around 30% in rural areas. Eastern regions of the country are particularly disadvantaged, with a coverage of less than 35% of households (EU average is 86%) (The share of households enjoying high-speed Internet connections is an indicator measuring EU's progress towards sustainable development goal 9 (concerning industry, innovation, infrastructure). In the countries of the European Union, 59.3% of households had a fixed, very-high-capacity network (VHCN) connection in 2020. While it constitutes a significant progress compared with the situation several years ago (e.g., the figure for 2013 was 15.6%), over 40% of households in the EU still do not enjoy such connectivity, and access varies in different income categories and locations. For example, the share of rural households with fixed VHCN connection stood at 27.8% across the EU [15] (p. 205)). The selected indicators are presented in Table 1, along with others referred to in the following part of the article.

**Table 1.** Background indicators: Poland and the European Union (EU) compared.

| Indicator | Poland | EU |
|---|---|---|
| VHCN (very-high-capacity network) connection in 2020 | 60% | 60% |
| Share of adults (16–74) having at least basic digital skills in 2019 | 44% | 56% |
| Early leavers from education and training in 2020 | 5.4% | 9.9% |
| Gender pay gap in an unadjusted form (% of average gross hourly earnings of men) in 2019 | 8.5% | 14.1% |
| Gender employment gap in 2020 | 11.1% | 15.7% |
| Inactive population due to caring responsibilities in 2019 (% of inactive population aged 20 to 64)—men | 12.5% | 3.9% |
| Inactive population due to caring responsibilities in 2019 (% of inactive population aged 20 to 64)—women | 40.1% | 27.3% |

Source: [29], Eurostat (Statistics | Eurostat (europa.eu), Statistics | Eurostat (europa.eu, accessed on 30 August 2021), Statistics | Eurostat (europa.eu, accessed on 30 of August 2021), Statistics | Eurostat (europa.eu, accessed on 30 of August 2021), Statistics | Eurostat (europa.eu, accessed on 30 of August 2021).

Living in areas with workplaces and schools lagging behind in digitalization puts people in a disadvantaged position, the same as not having the required level of digital skills to make use of the technology that is available. Due to the COVID-19 outbreak, the digital skills gap that had already existed before the pandemic has been accentuated, and new inequalities are emerging. In 2019, 56% of 16 to 74-year-old people in the EU had at least basic digital skills, a bit fewer women (54%) than men (58%) [15] (p. 117). This share for Poland is much lower (44%) and has been rising very slowly over the past few years. There is also a small male advantage in this indicator for Poland (46% vs. 43%). According to a poll carried out in March 2020, 76% of adults in Poland use the Internet at least once a week, others less frequently or not at all [30].

*5.2. COVID-19 and Educational Inequalities*

A poll carried out in January 2021 by the Public Opinion Research Center [30] in Poland among parents of school age children (in primary or secondary school), concerning assessment and experiences with online education, demonstrated that about 20 per cent of Poles had a child of school age learning online at the time of research (mostly one (70 per cent) or two (25 per cent) and three or more in 5 per cent of cases). In general, parents considered online education as being of lesser quality than in-person learning (89 per cent of parents, with 62 per cent deeming it much worse). When asked about problems involved in online learning, 37 per cent of parents pointed to excessive involvement and burden placed on themselves (to put this number into context, almost twice as many mentioned a lack of contact or limited contact with peers, too much time spent in front of the computer and lack of physical activity). The burden placed on parents was mentioned more often by parents with higher levels of education. This data might point to the fact that better educated (and probably more affluent) parents tend to spend more time attending to the education of their children. These children's advantage over their less well-off peers also stems from better access to modern technologies, experienced by affluent households, especially in urban areas.

According to the same poll, the majority of pupils had online meetings with teachers (82 per cent on average), particularly in the first three grades of primary schools (90 per cent), whereas others were sent assignments and educational materials to do on their own. The type of online learning (live meetings or assignments) seems particularly relevant from the standpoint of prospective burden for parents, as early-grade pupils are still developing their reading and writing skills and are mostly unable to work on their assignments on their own, which was expected from every tenth pupil of grades 1 to 3. Indeed, parents in this study declared that only 30 per cent of these pupils learned on their own, whereas for grades 4–8, this percentage was 42 (and 51 per cent on average). More involvement on the part of family was required when children were sent assignments. The parents of primary school children more often than others pointed to problems with learning encountered by their offspring.

In view of the aforementioned data, it may be hypothesized that the period of online education under lockdown conditions has contributed to the deepening of educational inequalities in Poland, with pupils with better access to modern technologies and of affluent and better educated parents receiving more attention at home and those from poorer backgrounds falling behind. The latter issue comes up in interviews with representatives of educational institutions, reporting the problem of pupils "disappearing from the educational system" who did not participate in online classes, did not complete their assignments and the contact with whom was hindered. In a representative study of staff of educational facilities, 48% of teachers from primary and secondary schools reported that at least one pupil did not participate in online education (58% of teachers in vocational schools). Another problem reported in the study were pupils who logged on to the system but did not participate in classes [31,32].

*5.3. COVID-19 and Economic and Gender Inequalities*

Has the increased burden on parents due to school closures, reported in public opinion surveys, contributed to reduced participation of parents (particularly women) in the labor market? Has it curtailed households' resources in an unequal way, potentially contributing to increased inequalities? On the other hand, have changes on the labor market (such as layoffs or reduced working hours) affected men and women in a different way?

In general, households' resources have been significantly affected by the pandemic, with layoffs in many sectors of the economy, reduced working time and, subsequently, wages. In the European Union, GDP per capita dropped substantially in 2020 compared with the previous year, going hand in hand with a fall in investment and employment and a considerable increase in the share of people not in education, employment and training [33]. In the second quarter of 2020, the EU's GDP dropped by 11.2%, which resulted in an annual

drop of real GDP per capita by 6.2% in 2020 compared with 2019. Industrial production experienced an annual decrease of 8.0% in 2020 compared with 2019 [33] (p. 35). Most of the employment indicators in the EU showed improvements in the third and fourth quarters of 2020 and—as of the moment of writing this article—the economy has been recovering with an expected growth of 4.2% in 2021 (according to the European Commission's latest Economic Forecast).

The pandemic's impact on the Polish economy was less severe than in many other European countries. For example, the GDP dropped by 8.9% in the second quarter of 2020 (the average for the EU was 11.1%) but soon recovered, reaching 7.7% in the third quarter, falling again and oscillating around 0–2% since then [34], close to the EU average. The monthly unemployment rate rose slightly after March 2020 when it stood at 2.9% but did not exceed 3.4% until the end of the year (and 4.0% in the first half of 2021) and remained a few percentage points below the EU average (around 7% in various months of 2020).

As for the pandemic's impact on labor force participation, it seems that it has been moderate so far. In Poland, there was a slight decline in the proportion of people outside the labor force (by 0.4 pp) down to 29% in 2020. There is, however (and has been for a long time), a significant gap between men and women: 21.7% of men are inactive, compared with 36.4% of women. In the European Union, on the other hand, the share of people outside the labor force increased slightly from 26.6% in 2019 to 27.1% in 2020, for the first time after 17 years of downward trend. The share of women outside the labor force has been consistently higher than that of men since Eurostat started the time series in 2002. However, the gender gap in the EU declined since 2002 (when it was 16.7 percentage points), reaching its lowest point in 2020 (10.7 pp).

Caring responsibilities are the main reason for inactivity among women. In general, Poland has a fairly large inactive population due to caring responsibilities (30.9% versus 18.7% for the whole EU). (According to the definition used in the EU Labour Force Survey (EU-LFS), the economically inactive population comprises individuals that are not working, not actively seeking work and not available to work even if they have found a job. Therefore, they are neither employed nor unemployed and considered to be outside the labour force, for example, because they are enrolled in education, caring for a family member, retired or because of illness or disability) There is, however, a particularly large gender gap for this indicator in Poland, with over 40% of inactive women aged 20–64 being outside the labour force for this reason and 12.5% of men.

Those women who are part of the labour force, despite their higher on average educational attainment, are still paid less, which is evidenced by the persistent gender pay gap. Women's gross hourly earnings in Poland are 8.5% lower than men's. While significant, this difference is smaller than the average for the EU (14.1%). Eurostat's analyses demonstrate that women in the whole EU are overrepresented in low pay sectors and underrepresented in well-paid sectors [15]. Due to the gender pay gap, as well as interrupted and shorter working lives, they earn less over their lifetime than men. Lower employment rates and higher involvement in caring responsibilities aggravates women's risk of poverty and social exclusion, especially in old age. As of the moment of writing this article, data were not available to assess any changes in the gender pay gap between 2019 and 2020.

*5.4. The Gendered Impact of School and Childcare Closures*

Women's employment is shaped not only by broader inequality patterns in the labour market but also partner dynamics and the way parents divide paid and unpaid work and care-related institutions (e.g., schools and childcare systems [25] (p. 5)). As evidenced by the data on labour force participation, women are more likely than men to remain outside the labour market due to caring responsibilities. As mothers, they are more likely than fathers to be stay-at-home parents, work part-time or adjust their work hours to accommodate family needs. In 2020, 73.8% of women and 92.1% of men aged 25–54 living in a couple and with children were employed in the EU. In Poland, these shares were

75.8 and 95, respectively. The employment rate for women being in a couple and without children was 84.3%, while that for men was 92.1%. For all women (whether in a couple or not) employment rate when they had children was 74.1% in Poland (0.4% more than in 2019) and 93.5% for men (no change between 2019 and 2020). Therefore, having children affects men and women differently, with more men entering the labor market and more women not working in this case, especially if they have more children and a lower level of education. Relevant data are presented in Table 2 (for adults in a couple).

**Table 2.** Employment rate by sex, educational attainment and household composition (% of total population aged 25–54), 2020 [1].

| Indicator | Poland | | EU | |
|---|---|---|---|---|
| | **Men** | **Women** | **Men** | **Women** |
| Adult living in a couple without children, all ISCED 2011 levels | 92.1 | 84.3 | 90.4 | 81.1 |
| Adult living in a couple with children, all ISCED 2011 levels | 95.0 | 75.8 | 92.1 | 73.8 |
| Less than primary, primary and lower secondary education (levels 0–2), without children | 81.8 | 61.1 | 79.7 | 57.5 |
| Less than primary, primary and lower secondary education (levels 0–2), with children | 86.8 | 45.7 | 81.6 | 46.0 |
| Upper-secondary and post-secondary non-tertiary education (levels 3 and 4), without children | 89.5 | 76.4 | 90.6 | 80.8 |
| Upper-secondary and post-secondary non-tertiary education (levels 3 and 4), with children | 93.8 | 64.3 | 92.9 | 72.0 |
| Tertiary education (levels 5–8), without children | 97.0 | 94.2 | 94.0 | 88.5 |
| Tertiary education (levels 5–8), with children | 97.6 | 86.9 | 96.1 | 84.6 |

Source dataset: https://ec.europa.eu/eurostat/databrowser/view/LFST_HHEREDTY__custom_1273495/default/table?lang=en (accessed on 8 September 2021). [1] Data in this table refer only to adults in a couple.

Contrary to some data from online surveys suggesting that women in some countries, including Poland, were more likely than men to stop working at the beginning of the pandemic [35], Eurostat's data for the whole year of 2020 demonstrate that between 2019 and 2020 there was a small, up to a percentage point increase in the employment rate for women with children across all levels of education (and a significant rise for women with children with the lowest level of education—from 35.8% to 42.3%), while the employment of women without children did not change or decreased by half percentage point across all levels of education.

According to the Labor Force Survey, the (harmonized) unemployment rate for women was similar to that of men in Poland at around 3.5% and 3.4%, respectively, in July 2021, not changing more than a few decimal percentage points throughout the pandemic [36]. Part-time employment is not very popular in Poland, with only 5.9% of employed persons working in this way (18.2% in the whole EU). For men and women, these percentages are 3.4% and 8.9%, respectively (8.4% and 29.7% in the EU), with slight declines between 2019 and 2020 (by 0.2 percentage points) [37].

*5.5. Gender Roles Inacted during the Pandemic*

The coronavirus disease pandemic has forced society to confront a new work–family situations for most families. The primary responsibility for home and family has traditionally been assigned to women. Women's dedication to the family sphere and, more

specifically, caregiving responsibilities associated with raising children, contrasts with paid work as the central domain for men with their dedication to the role of "good provider". When asked about the preferred model of family life, 58% of adult Poles opt for one in which there is an equal division of labor at home and both partners devote the same amount of time for employment in the labor market (according to a representative public opinion poll conducted in October 2020; support for this model has increased by 12 percentage points since the previous survey in 2013) [38]. The traditional bread-winner model is supported by only 14% of respondents; every fifth one (20%), though, is in favor of husband being more devoted to paid employment and the wife taking care of children and household apart from working in the labor market. Despite a widespread support for partnership in family life, there exists a fairly significant share of Poles who support the notion of a woman being at least more, if not solely, responsible for children and household chores. On the other hand, preferences do not always translate into practice, and about one third of Poles do not live according to their preferences. As evidenced by the data in Table 2, women are indeed more likely to leave paid employment when they have children. Has the pandemic changed people's preferences and the actual division of labor at home and in the labor market, or is there evidence to the contrary? There is limited data to resolve this issue, however, certain preliminary conclusions can be drawn from public opinion polls on the general impact of the pandemic on people's lives.

According to an online poll carried out in May 2020 [39], women experienced more often than men a sense of loneliness during lockdown (43% compared with 33%), which can probably be attributed to the longer life expectancy of females in Poland and the fact that there are more women living alone than men. On the other hand, women more often than men missed the opportunity to separate themselves from other family members (19% compared with 14%). The authors of this research speculate that the burden of caring for family members might have been experienced more acutely by women, since they are more involved in household chores on a day-to-day basis, and it was more difficult to get away from them during the period of social isolation. This conclusion is further confirmed by the fact that women, much more often than men (30% versus 19%), deemed the closing of educational institutions (daycares, schools and universities) as one of the most acute restrictions during the pandemic. These differences point to differing responsibilities with respect to the supervision of children's activities.

## 6. Summary of Findings

The data analyzed in this article, as well as the research conducted so far, demonstrate that the pandemic has had a profound impact on the educational system in Poland. It is very likely that the pandemic has deepened educational inequalities. This claim seems justified particularly in view of the data on the prevalence and use of modern technologies. Digitalization in Poland (in terms of access to high-speed internet connections) is not universal, with rural areas and eastern regions of the country being relatively disadvantaged on some indicators. On the other hand, the ability to use modern technology, even if it is available, is even lower, standing at 44% of the adult population with at least basic digital skills (with a male advantage of 3 percentage points). Moreover, preliminary and mostly qualitative research carried out in Poland suggests that homeschooling was not universal, with (as was pointed to in interviews with teachers) some children being left to their own devices, unassisted (for various reasons, including parents involved in paid employment, neglect, etc.).

The data analyzed herein do not support the claim that women in Poland experienced particular economic difficulties in terms of curtailed labor market participation as a result of the COVID-19 pandemic's outbreak. The share of men and women active on the labor market is unequal to the advantage of men, both in Poland and in most other European countries, reflecting varying enrolments in tertiary education (usually to the advantage of women), discrimination on the labor market (usually against women) and different gender roles. Consequently, the gender employment gap stands at 11.1% in Poland, a few

percentage points less than the average for the EU (15.7%). (Gender employment gap is defined as the difference between the employment rates of men and women aged 20 to 64. The employment rate is calculated by dividing the number of people aged 20 to 64 in employment by the total population of the same age group. The indicator is based on the EU Labour Force Survey (EU-LFS)). It should be stressed, though, that as many as 36.4% of women in Poland are economically inactive and caring responsibilities are by far the main reason for inactivity among women. These facts alone might have alleviated the negative effects of increased care-related needs for children due to restrictions on schooling and daycare. Contrary to evidence from other countries [24,35], women in Poland have not reduced their (already lower than in many other EU countries) labor market participation to meet increased caregiving needs, and for women with the lowest level of education, the opposite is true—a significant rise in the activity rate was registered in labor market surveys. The latter might be due to the fact that an increased demand for low-skilled labor in some sectors of the economy, combined, perhaps, with economic difficulties experienced by some families, pushed more women into the labor market. Unemployment rate remained low during the pandemic in Poland, with no significant increase neither for women, nor for men. There was neither an increase nor decrease in part-time employment, which is, in general, less common than in other EU countries.

The pandemic is an ongoing process, the effects of which are yet to be studied and many are probably not yet reflected in official statistics [40]. On the other hand, economic impacts extend below the study of the selected indicators being the focus of this article. Bearing these caveats in mind, we hypothesize that while having a profound impact on the educational system, the pandemic in Poland has not changed gender structures in a significant way—at least in terms of a transformation of the care burden experienced mostly by women. It has simply contributed to preserving the status quo. COVID-19, with its lockdowns and school closures, hit a country characterized by a significant gender employment gap where a large share of women remain outside the labor market due to the care needs of family members. We hypothesize that on the societal level, this gap "absorbed" most of the shock delivered by the pandemic in terms of increased care needs.

## 7. Discussion

Eurostat's official statistics examined in this article show some of the early impacts of the COVID-19 pandemic in the sphere of education, labor market and gender equality. The 2021 edition of a report by the Eurostat, based on these and other data and monitoring progress towards the SDGs in an EU context states that the pandemic has made achieving the 2030 Agenda and the SDGs even more challenging than before (both for the EU and globally) [15] (p. 9). Apart from the economy and the labor market, COVID-19 has markedly slowed the average progress in most other areas covered by sustainable development goals, including education, gender equality and reducing other inequalities, "where 2020 data show a clear deterioration for individual indicators".

Are these conclusions pertinent to the situation of Poland? In our research, we aimed to find out in what way the pandemic has affected the education system in Poland and, in particular, whether there was evidence that the pandemic has deepened educational inequalities, as elsewhere across the world, exacerbating existing gaps in schooling. There is some evidence that this has indeed happened in Poland in the sphere of education, as summarized in the preceding paragraph. As is evidenced in the data, households' resources (their economic, social and cultural capital) have been negatively affected during the pandemic, potentially contributing to gaps in schooling (in what way though remains to be studied). Remote instruction imposed learning challenges but physical absence from the school environment also compromised the physical and mental well-being of children and adolescents. What raises concerns is unequal access to technological resources, as well as the wide variability in the quality and implementation of online programs along class lines but also varying depending on rural versus urban places of residence.

Gender equality is one of the UN's sustainable development goals, with a stress placed on the full participation of men and women in economic and political life. Therefore, our second research question was about the pandemic's impact on women's and men's position on the labor market. Around the world, the COVID-19 pandemic strongly affects sectors with high shares of female employment. Another channel of impacts of COVID-19 is an increased need for childcare for mothers due to school closures. The hypothesis that the pandemic has affected women's situation on the labor market in a more negative way than men's situation has not been confirmed for Poland. From the sustainability perspective, women's situation on the labor market had already been relatively unfavorable before the pandemic (high inactivity rates, especially due to caring responsibilities), and has remained so since the pandemic's outbreak (except for women with the lowest level of education). As hypothesized in the preceding paragraph, on the societal level, the gap between men and women in Poland in terms of inactivity rates "absorbed" most of the shock delivered by the pandemic in terms of increased care needs.

Lastly, we asked about the pandemic's impact on gender and parental roles in Poland and whether changes have occurred in social attitudes with the potential to ease the burden that the pandemic has placed on children's schooling. Although the majority of Poles support the model of equal division of labor within the household, there exists a fairly significant share who support the notion of a woman being at least more, if not solely, responsible for children and household chores. On the other hand, preferences do not always translate into practice, and about one third of Poles do not live according to their preferences, with women being more likely to leave paid employment when they have children. There is too limited data to reach any definite conclusions on whether and in what way the pandemic has changed gender and parental roles in Poland. However, the data on more women than men missing the opportunity to separate themselves from other family members and considering the closing of educational institutions (daycares, schools and universities) as one of the most acute restrictions during the pandemic might point to the fact that women remain more involved in household chores on a day-to-day basis.

As subsequent waves of the new variants of the coronavirus sweep across the world, the full effects that the pandemic has had on societies remain to be studied. This study has offered a preliminary insight into the situation of a country characterized by a significant gender employment gap, a considerable gender pay gap and moderate levels of digitalization.

**Author Contributions:** Conceptualization, M.K.-A. and Ł.A.; methodology, M.K.-A. and Ł.A.; validation, M.K.-A. and Ł.A.; formal analysis, M.K.-A.; investigation, M.K.-A.; resources, M.K.-A.; writing—original draft preparation, M.K.-A. and Ł.A.; writing—review and editing, M.K.-A. All authors have read and agreed to the published version of the manuscript.

**Funding:** This research received no external funding.

**Institutional Review Board Statement:** Not applicable (the study did not require Institutional Review Board's Statement nor approval, because it uses only publicly available data; it is unobtrusive research).

**Informed Consent Statement:** Not applicable.

**Data Availability Statement:** Data supporting reported results can be found at https://ec.europa.eu/eurostat/data/database (accessed on 20 September 2021).

**Conflicts of Interest:** The authors declare no conflict of interest.

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
