# Peer review of "Gender and Educational Inequalities during the COVID-19 Pandemic: Preliminary Insights from Poland"

_sustainability, doi:10.3390/su132212403_

Round 1

Reviewer 1 Report

The work is interesting and addresses a topical issue.

However, compared to the extensive section in the introduction/theoretical framework, the methodological framework is scarce. It does not address when the data was collected, why the instruments were selected, what categories these instruments develop, data analysis, etc. This section is very poor and I think it would be essential to address it.

The same goes for the discussion. Compared to the extensive body of content of the first section and the results, this section is very scarce. The research questions should be addressed or answered.

Author Response

Thank you for your comments, they have helped us improve our article, rethink our goals and clarify our arguments.

  • “(…) compared to the extensive section in the introduction/theoretical framework, the methodological framework is scarce. It does not address when the data was collected, why the instruments were selected, what categories these instruments develop, data analysis, etc. This section is very poor and I think it would be essential to address it.” This comment from the reviewer is appreciated. We added to the methodological section explaining the time frame which was chosen and the reasons for focusing on particular data.
  • “ Compared to the extensive body of content of the first section and the results, this section is very scarce. The research questions should be addressed or answered”. We added to the discussion addressing our research questions in more detail.

Reviewer 2 Report

The subject of the work is very interesting, the theoretical framework is correctly configured, however the work method and the empirical part lack entity.

A descriptive analysis of three reports, I consider that they do not provide sufficient entity to the article since the authors only dedicate themselves to making a description of the data from different sources without providing in this article more information than that which appears in said reports.

I consider a complete new structuring of the methodological part should be done since the article does not contribute anything new that is not contemplated by the cited sources.

It is recommended to carry out a comparative analysis or a content analysis of the reports in a more detailed way, establishing relationships with the different points of the article.

Author Response

  • “(…) the work method and the empirical part lack entity. A descriptive analysis of three reports, I consider that they do not provide sufficient entity to the article since the authors only dedicate themselves to making a description of the data from different sources without providing in this article more information than that which appears in said reports”. In our article, we occasionally refer to reports by the Eurostat (mostly Sustainable development in the European Union), however, most of the data have been drawn directly from Eurostat’s database. We added links to the database where relevant. Our contribution to understanding post-Covid Polish society does not consist in gathering new data, but in drawing on existing statistics to reach novel conclusions. We believe that we have managed to this (the aforementioned conclusions do not appear in the aforementioned reports, they constitute our original input).
  • “I consider a complete new structuring of the methodological part should be done since the article does not contribute anything new that is not contemplated by the cited sources. It is recommended to carry out a comparative analysis or a content analysis of the reports in a more detailed way, establishing relationships with the different points of the article.” As mentioned above, our contribution to understanding post-Covid Polish society does not consist in gathering new data, but in drawing on existing statistics to reach novel conclusions. We analyse existing statistics, which are obviously discussed in various sources, however, we use them following the theoretical framework outlined in the first paragraphs and so as to answer our research questions. We understand that deficiencies in our methodological section and in discussion might have obscured the arguments and conclusions presented in our paper. We changed these sections, clarifying why we use particular data and where they come from and addressing each research question one by one. We do not carry out comparative analysis nor content analysis of any published reports which happen to use the same data as we use because Eurostat’s objectives when publishing them are different from the objectives of our paper which are stipulated in our research questions. Nor are we interested in the assumptions lying behind the presentations of official statistics, nor other goals of content analysis.

Thank you for your comments, they have helped us rethink our goals and clarify our arguments.

Reviewer 3 Report

Thank you for the opportunity to comment on the evaluated contribution Gender and Educational Inequalities During the Covid-19 Pandemic: Preliminary Insights From Poland.

I consider that the topic of the study is really cutting-edge, being the research problem a very relevant one; the literature review is adequate and up to date, the theoretical framework is clearly and correctly defined, supported by well-founded assumptions; I also value positively the methodological part. The literature cited is relevant to the study (could be increased).

The topic of the article is current and suitable for solution. I recommend editing the abstract. I recommend clearly formulating the aims of the article and adding them to the abstract. It would be appropriate to add to the abstract the basic specific methods that were used in creating the paper.

Chapter 4. Methods I recommend to elaborate more in terms of methodological basis and give examples of use in the article.

I find the work interesting and can serve as a reference for further studies.

Author Response

Thank you for your comments, they have helped us rethink our goals and clarify our arguments.

  • “I recommend editing the abstract. I recommend clearly formulating the aims of the article and adding them to the abstract. It would be appropriate to add to the abstract the basic specific methods that were used in creating the paper.” We edited the abstract and added more information on the methods that were used.
  • “Chapter 4. Methods I recommend to elaborate more in terms of methodological basis and give examples of use in the article”. We analyze existing statistics and use them following the theoretical framework outlined in the first paragraphs and so as to answer our research questions. We added to the methodological section, explaining our objectives.

Round 2

Reviewer 1 Report

Dear authors,
I consider that the proposed modifications have been implemented successfully.
My congratulations.
All the best,

Reviewer 2 Report

The authors have improved some of the proposed questions, questions that I think have improved the quality of the article and have given consistency to the methodological framework.